# Structural insights into the orthosteric inhibition of P2X receptors by non-ATP analog antagonists

**Danqi Sheng[1†], Chen-Xi Yue[2†], Fei Jin[1], Yao Wang[1], Muneyoshi Ichikawa[3], Ye Yu[2], Chang-Run Guo[2]\*, Motoyuki Hattori[1]\***

[1]State Key Laboratory of Genetic Engineering, Shanghai Key Laboratory of Bioactive Small Molecules, Collaborative Innovation Center of Genetics and Development, Department of Physiology and Neurobiology, School of Life Sciences, Fudan University, Shanghai, China; [2]School of Basic Medicine and Clinical Pharmacy, China Pharmaceutical University, Nanjing, China; [3]State Key Laboratory of Genetic Engineering, Department of Biochemistry and Biophysics, School of Life Sciences, Fudan University, Shanghai, China

**\*For correspondence:**
gcr@cpu.edu.cn (C-RG);
hattorim@fudan.edu.cn (MH)

[†]These authors contributed equally to this work

**Competing interest:** The authors declare that no competing interests exist.

**Abstract** P2X receptors are extracellular ATP-gated ion channels that form homo- or hetero-trimers and consist of seven subtypes. They are expressed in various tissues, including neuronal and nonneuronal cells, and play critical roles in physiological processes such as neurotransmission, inflammation, pain, and cancer. As a result, P2X receptors have attracted considerable interest as drug targets, and various competitive inhibitors have been developed. However, although several P2X receptor structures from different subtypes have been reported, the limited structural information of P2X receptors in complex with competitive antagonists hampers the understanding of orthosteric inhibition, hindering the further design and optimization of those antagonists for drug discovery. We determined the cryogenic electron microscopy (cryo-EM) structures of the mammalian P2X7 receptor in complex with two classical competitive antagonists of pyridoxal-5'-phosphate derivatives, pyridoxal-5'-phosphate-6-(2'-naphthylazo-6'-nitro-4',8'-disulfonate) (PPNDS) and pyridoxal phosphate-6-azophenyl-2',5'-disulfonic acid (PPADS), and performed structure-based mutational analysis by patch-clamp recording as well as molecular dynamics (MD) simulations. Our structures revealed the orthosteric site for PPADS/PPNDS, and structural comparison with the previously reported apo- and ATP-bound structures showed how PPADS/PPNDS binding inhibits the conformational changes associated with channel activation. In addition, structure-based mutational analysis identified key residues involved in the PPNDS sensitivity of P2X1 and P2X3, which are known to have higher affinity for PPADS/PPNDS than other P2X subtypes.

## eLife assessment

This study by Sheng and colleagues provides **valuable** insights into the mechanism of competitive inhibitors of P2X receptors. The structural and functional evidence supporting the subtype specificity of pyridoxal-5'-phosphate derivatives is **compelling** and provides information for designing drugs that selectively target different subtypes of P2X receptor channels. The work will be of interest to biochemists, structural biologists, and pharmacologists.

## Introduction

ATP not only serves as a cellular energy currency but also plays a key role in signal transmission for cellular stimulation between cell surface receptors (*Burnstock, 1972*). P2X receptors are the family of cation channels activated by extracellular ATP and are widely expressed in the mammalian nervous, respiratory, reproductive, and immune systems (*Valera et al., 1994*; *Chen et al., 1995*; *Khakh and North, 2006*; *Burnstock et al., 2014*). There are seven subtypes (P2X1-P2X7) in the family, each of which plays distinct roles in physiological and pathophysiological functions via homo- or heterotrimerization (*Brake et al., 1994*; *North, 2002*). In recent years, there has been growing interest in the development of drugs targeting the P2X family due to its involvement in various physiological and pathological conditions, and various antagonists have been developed (*Illes et al., 2021*). Some have progressed to clinical trials (*Illes et al., 2021*), and Gefapixant, a P2X3 receptor antagonist for chronic cough, was already on the market after the clinical study (*McGarvey et al., 2022*).

ATP analogs are most common among competitive inhibitors for P2X receptors; however, they are generally unsuitable for in vivo applications due to their relatively low specificity, which may result in off-target toxicity. This issue arises because the human body contains numerous ATP-binding proteins. Therefore, a non-ATP analog P2X inhibitor would be a more promising target to develop and optimize, and pyridoxal phosphate-6-azophenyl-2',5'-disulfonic acid (PPADS) and its analog pyridoxal-5'-phosphate-6-(2'-naphthylazo-6'-nitro-4',8'-disulfonate) (PPNDS) are such classical non-ATP analog P2X inhibitors, namely, pyridoxal phosphate derivatives (*Ralevic and Burnstock, 1998*; *Lambrecht et al., 2002*; *Jacobson et al., 2002*; *Kaczmarek-Hájek et al., 2012*). PPNDS and PPADS belong to the class of competitive antagonists that selectively inhibit P2X receptors (*Huo et al., 2018*), and P2X receptors are known to exhibit variable sensitivity to PPADS/PPNDS depending on the species and the specific subtype (*Kaczmarek-Hájek et al., 2012*; *Michel et al., 2008*; *Donnelly-Roberts et al., 2009*; *Buell et al., 1996*; *Garcia-Guzman et al., 1997*; *Jones et al., 2000*). It is noteworthy that P2X1 and P2X3 receptors show relatively high sensitivity to PPADS, but P2X2 and P2X7 receptors show only moderate sensitivity, and P2X4 receptors are insensitive to PPADS (*North and Surprenant, 2000*).

Several attempts have been made to optimize pyridoxal phosphate derivatives as P2X antagonists (*Brown et al., 2001*; *Cho et al., 2013*; *Kim et al., 2001*; *Jung et al., 2013*). For example, the introduction of bulky aromatic groups at the carbon linker in PPADS was attempted to improve in the subtype specificity profiles, potentially opening up new avenues for targeted drug development and therapeutic intervention (*Cho et al., 2013*). Despite the recent significant increase in structural information on P2X receptors (*Kawate et al., 2009*; *Hattori and Gouaux, 2012*; *Kasuya et al., 2016*; *Karasawa and Kawate, 2016*; *Mansoor et al., 2016*; *Kasuya et al., 2017*; *Wang et al., 2018*; *Li et al., 2019*; *McCarthy et al., 2019*), the lack of structural information on P2X receptors in complex with pyridoxal phosphate derivative inhibitors has hampered their rational optimization for drug discovery targeting P2X receptors.

In this work, we determined the cryogenic electron microscopy (cryo-EM) structures of the panda P2X7 (pdP2X7) receptor in complex with PPADS and PPNDS. The structures revealed the orthosteric binding site for these pyridoxal phosphate derivatives. Structural comparison with the previously determined apo- and ATP-bound P2X7 receptors (*Kasuya et al., 2017*; *McCarthy et al., 2019*) showed PPADS/PPNDS-dependent structural rearrangement at the orthosteric binding site for channel inactivation. Further mutational analysis by electrophysiological recording identified key residues of human P2X1 (hP2X1) and P2X3 (hP2X3) that show high sensitivity to pyridoxal phosphate derivative inhibitors.

## Results

### Structural determination and functional characterization

To gain insight into the mechanism of P2X receptor inhibition by pyridoxal phosphate derivatives, we used giant panda (*Ailuropoda melanoleuca*) P2X7, whose structures in complex with allosteric modulators have been reported (*Karasawa and Kawate, 2016*). Notably, pdP2X7 shares 85% identity with the human P2X7 receptor and exhibits high and stable expression profiles suitable for structural studies (*Karasawa and Kawate, 2016*). We performed whole-cell patch-clamp recordings using human embryonic kidney 293 (HEK293) cells transfected with full-length pdP2X7. The application of 10 µM PPNDS and 100 µM PPADS blocked approximately 50% of the ATP-dependent currents from pdP2X7 (*Figure 1—figure supplement 1*). This result correlates well with the properties of P2X7

receptors moderately inhibited by PPADS/PPNDS (*Donnelly-Roberts et al., 2009*; *Lambrecht et al., 2000*).

We then expressed and purified the previously reported crystallization construct of pdP2X7~cryst~ (*Karasawa and Kawate, 2016*). The purified pdP2X7~cryst~ was reconstituted into lipid nanodiscs, mixed with PPNDS and PPADS, separately, and subsequently subjected to single-particle cryo-EM (*Figure 1—figure supplements 2–5*). The structures of pdP2X7 in the presence of PPNDS and PPADS were determined at 3.3 Å and 3.6 Å, respectively (*Table 1*).

The overall structures are similar and show the trimeric architecture of P2X receptors, consisting of the extracellular domain and two transmembrane (TM) helices, with each protomer resembling the dolphin shape, consistent with the previously reported P2X structures (*Sheng and Hattori, 2022*; *Figure 1*, *Figure 1—figure supplement 6*). More importantly, we identified the residual EM densities at the agonist binding site that fit into the shape of PPNDS and PPADS (*Figure 1*). It should be noted that while the pyridoxal phosphate groups in PPNDS and PPADS are shared, the naphthylazo group of PPNDS is significantly bulkier than the azophenyl group of PPADS. This difference facilitated our assignment of compound binding poses corresponding to each EM density (*Figure 1*). Consistent with antagonist binding, the TM domain adapts to the closed conformation of the channel, as similarly observed in the previously reported closed state structures of P2X7 receptors (*Karasawa and Kawate, 2016*; *McCarthy et al., 2019*).

## Orthosteric binding site

In the PPNDS-bound and PPADS-bound structures, PPNDS and PPADS molecules bind to essentially the same orthosteric site, consistent with both PPNDS and PPADS being pyridoxal phosphate derivatives (*Figures 2 and 3*, *Figure 2—figure supplement 1*). Furthermore, the residues involved in PPNDS and PPADS largely overlap with the residues at the ATP-binding site in the previously reported ATP-bound P2X7 structure (*Figure 4*; *McCarthy et al., 2019*). Many of the residues are highly conserved among P2X receptors and have been shown to be crucial for P2X activation (*Hattori and Gouaux, 2012*; *Jiang et al., 2000*; *Ennion et al., 2000*; *Bodnar et al., 2011*), which is consistent with both of them being competitive inhibitors.

In both structures, the pyridoxal phosphate group has extensive interactions with the receptor. In contrast, the other parts of the compounds, a naphthylazo group with two sulfonic acid groups and a nitro group (PPNDS) and an azophenyl group with two sulfonic acid groups (PPADS), have fewer interactions (*Figures 2 and 3*).

The phosphate group of PPNDS and PPADS interacts directly with the side chain of Arg294 and possibly also with Lys145, possibly via a water molecule (*Figures 2B and 3B*, *Figure 2—figure supplement 1*). However, it should also be noted that it is difficult to conclude the existence of the water molecule at this site due to the limited resolution of our structures. Furthermore, in the PPADS-bound structure, Lys64 mediates an additional interaction with the phosphate group. These extensive interactions between the phosphate group and the receptor resemble those with the phosphate groups of ATP (*Figure 4A*). Furthermore, the Asn292 and Lys311 residues are similarly involved in the interaction with the hydroxyl group of the pyridoxal part of PPNDS and PPADS (*Figures 2B and 3B*).

Interestingly, despite the structural differences between PPNDS and PPADS, the two common sulfonic acid groups form hydrogen bonds with the side chains of the same residues in the receptor, Lys66 and Gln143 (*Figures 2B and 3B*).

Finally, to verify the binding mode of the pyridoxal phosphate derivatives, we performed molecular dynamics (MD) simulations of the higher-resolution PPNDS-bound structure embedded in lipids (*Figure 2—figure supplement 2*). The overall structures were stable during the simulations, and PPNDS remained stably bound to the receptor (*Figure 2—figure supplement 2*).

## Structural comparison and inhibition mechanism

To gain insights into the mechanisms of the orthosteric inhibition of P2X receptors by PPNDS and PPADS, we superimposed our structures and the previously determined ATP-bound P2X7 structure onto the apo-state P2X7 structure (*Figure 5*). The PPNDS-bound structure and PPADS-bound structure are very similar, with 0.52 Å RMSD (root mean square deviation) values for the Cα atoms of 960 residues. Only the comparison with the higher-resolution PPNDS-bound structure is described in the following discussion.

**Table 1.** Cryogenic electron microscopy (cryo-EM) data collection, refinement, and validation statistics.

| | PdP2X7 w. PPNDS (EMD-36671) (PDB: 8JV8) | PdP2X7 w. PPADS (EMD-36670) (PDB: 8JV7) |
|---|---|---|
| Data collection and processing | | |
| Magnification | ×29,000 | ×29,000 |
| Voltage (kV) | 300 | 300 |
| Electron exposure (e–/Å$^2$) | 50 | 50 |
| Defocus range (µm) | –1.3 to –2.0 | –1.3 to –2.0 |
| Pixel size (Å) | 0.83 | 0.83 |
| Symmetry imposed | C3 | C3 |
| Initial particle images (no.) | 663,674 | 236,753 |
| Final particle images (no.) | 121,008 | 161,188 |
| Map resolution (Å) | 3.34 | 3.60 |
| FSC threshold | 0.143 | 0.143 |
| Map resolution range (Å) | 1.9–40.4 | 2.2–10.2 |
| Refinement | | |
| Initial model used (PDB code) | This study | This study |
| Model resolution (Å) | 3.34 | 3.60 |
| FSC threshold | 0.143 | 0.143 |
| Model resolution range (Å) | 1.9–40.4 | 2.2–10.2 |
| Map sharpening B factor (Å$^2$) | –50 | –150 |
| Model composition | | |
| Non-hydrogen atoms | 7245 | 7245 |
| Protein residues | 963 | 960 |
| Ligands | NAG:6, PPNDS:3 | NAG:6, PPADS:3 |
| B factors (Å$^2$) | | |
| Protein | 148.30 | 117.36 |
| Ligand | 189.65 | 146.40 |
| R.m.s. deviations | | |
| Bond lengths (Å) | 0.014 | 0.003 |
| Bond angles (°) | 1.257 | 0.561 |
| Validation | | |
| MolProbity score | 2.51 | 1.70 |
| Clashscore | 13.86 | 9.04 |
| Poor rotamers (%) | 3.70 | 1.22 |
| Ramachandran plot | | |
| Favored (%) | 93.50 | 97.17 |
| Allowed (%) | 6.39 | 2.83 |
| Disallowed (%) | 0.1 | 0 |

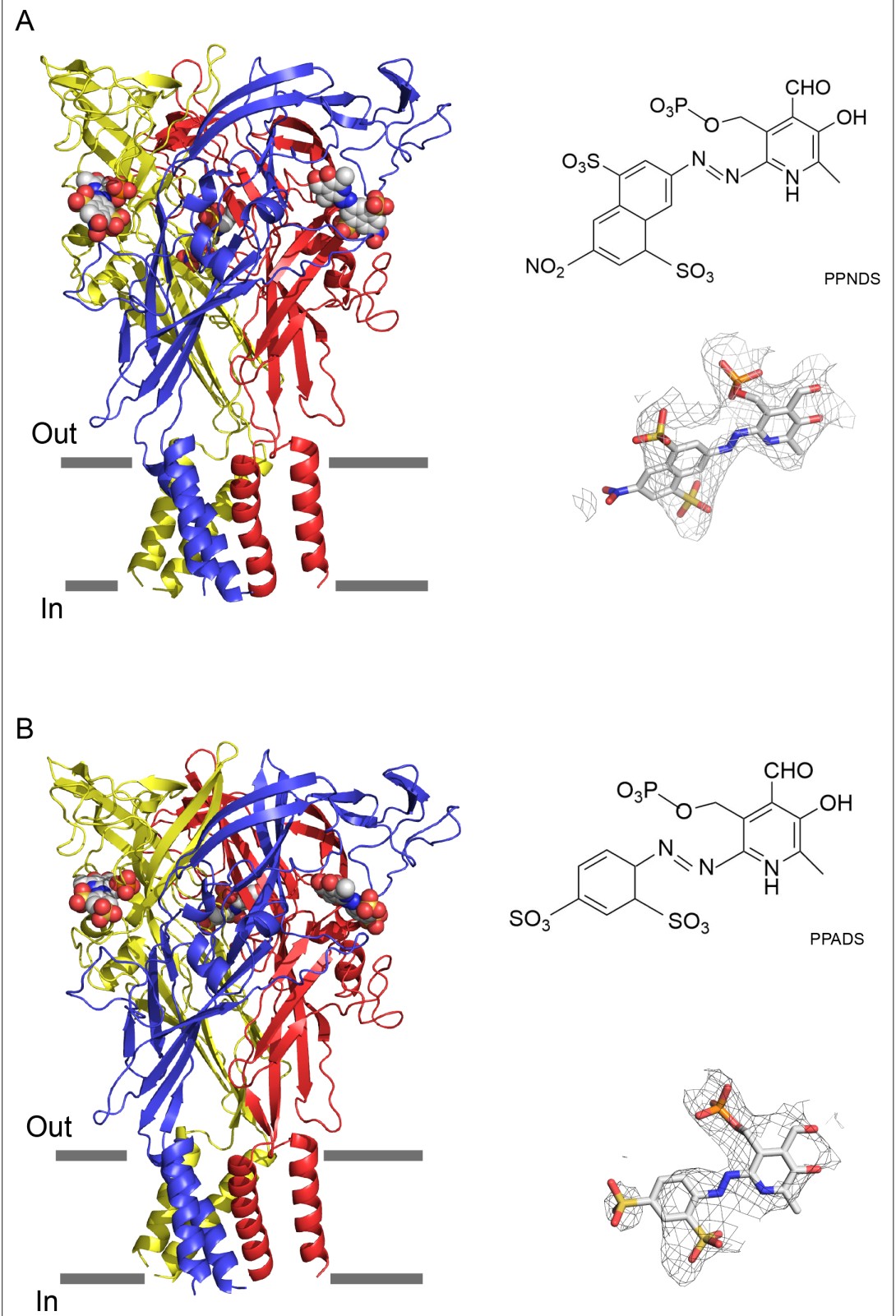

**Figure 1.** Cryogenic electron microscopy (cryo-EM) structures of pyridoxal-5'-phosphate-6-(2'-naphthylazo-6'-nitro-4',8'-disulfonate) (PPNDS)-bound and pyridoxal phosphate-6-azophenyl-2',5'-disulfonic acid (PPADS)-bound panda P2X7 (pdP2X7). The trimeric structures of PPNDS-bound (**A**) and PPADS-bound (**B**) pdP2X7, viewed parallel to the membrane. The PPNDS and PPADS molecules are shown as spheres. Each subunit of the trimers is colored

*Figure 1 continued on next page*

*Figure 1 continued*

blue, yellow, and red. The EM density maps contoured at 4.5σ and 3.5σ for PPNDS and PPADS are shown as gray mesh. The structural formulas of PPNDS and PPADS are also shown.

The online version of this article includes the following source data and figure supplement(s) for figure 1:

**Figure supplement 1.** Effects of pyridoxal-5'-phosphate-6-(2'-naphthylazo-6'-nitro-4',8'-disulfonate) (PPNDS) and pyridoxal phosphate-6-azophenyl-2',5'-disulfonic acid (PPADS) on panda P2X7 (pdP2X7) by patch-clamp recording.

**Figure supplement 1—source data 1.** Numerical data for *Figure 1—figure supplement 1C and D*.

**Figure supplement 2.** Cryogenic electron microscopy (cryo-EM) analysis of pyridoxal-5'-phosphate-6-(2'-naphthylazo-6'-nitro-4',8'-disulfonate) (PPNDS)-bound panda P2X7 (pdP2X7).

**Figure supplement 3.** Cryogenic electron microscopy (cryo-EM) data process for pyridoxal-5'-phosphate-6-(2'-naphthylazo-6'-nitro-4',8'-disulfonate) (PPNDS)-bound panda P2X7 (pdP2X7).

**Figure supplement 4.** Cryogenic electron microscopy (cryo-EM) analysis of pyridoxal phosphate-6-azophenyl-2',5'-disulfonic acid (PPADS)-bound panda P2X7 (pdP2X7).

**Figure supplement 5.** Cryogenic electron microscopy (cryo-EM) data process for pyridoxal phosphate-6-azophenyl-2',5'-disulfonic acid (PPADS)-bound panda P2X7 (pdP2X7).

**Figure supplement 6.** Dolphin model.

First, the activation of P2X receptors from the apo (closed) state to the ATP-bound (open) state is known to require motions of both the head and left flipper domains (*Lörinczi et al., 2012*; *Wang et al., 2017*; *Zhao et al., 2014*; *Figure 5A*). These motions are coupled to the movement of the lower body domain, which is directly connected to the TM domain for channel opening (*Figure 5A*).

In the PPNDS-bound structure, while we observed motion of the head domain similar to that in the ATP-bound structure, there was only a small structural change in the left flipper domain (*Figure 5A*). Consequently, there was no structural change in the lower body domain or associated gating motion of the TM domain (*Figure 5A*). In the ATP-bound structure, the three phosphate groups of ATP in the U-shaped conformation pushed down the left flipper (*Figures 4A and 5A*). In contrast, both PPNDS and PPADS possess only one phosphate group, so there was no corresponding downward movement in the left flipper domain (*Figures 2 and 5A*).

To summarize, the structural comparison indicates how PPNDS and PPADS inhibit the ATP-dependent activation of P2X receptors (*Figure 5B*). The binding of these competitive inhibitors may prevent the downward movement of the left flipper domain, which is required for channel opening (*Figure 5B*).

## Structure-based mutational analysis

To analyze the mechanism of P2X receptor binding to the pyridoxal-5'-phosphate derivatives, we performed structure-based mutational analysis by whole-cell patch-clamp recording of pdP2X7 (*Figure 6* and *Figure 6—figure supplement 1*). Most of the residues involved in PPNDS and PPADS binding overlap with the conserved residues involved in ATP binding (*Figure 4B*). Thus, we did not generate mutants targeting these residues (Lys64, Lys66, Asn292, Arg294, and Lys311), as such mutations are known to severely affect or abolish ATP-dependent gating of P2X receptors (*Hattori and Gouaux, 2012*; *Jiang et al., 2000*; *Ennion et al., 2000*; *Bodnar et al., 2011*). Instead, we aimed to mutate the residues surrounding the ATP-binding site, which differ among P2X receptor subtypes. Such residues may be important for the subtype-specific differences in the affinity of the pyridoxal-5'-phosphate derivative to P2X receptors. To design these mutants, we performed structural comparisons of our structures with the AlphaFold-based structural model of hP2X1 (*Jumper et al., 2021*; *Figure 6A*), which has high affinity for both PPNDS and PPADS (*North and Surprenant, 2000*; *Lambrecht et al., 2000*). In particular, we compared the residues surrounding the ATP-binding site in our structure (Gln143, Val173, Ile214, Gln248, and Tyr288) with those in hP2X1 (*Figure 4B*). Based on these comparisons, we mutated these residues to the corresponding amino acid residues in hP2X1 (Gln143Lys, Val173Asp, Ile214Lys, Gln248Lys, and Tyr288Ser) or to alanine (Gln143Ala, Gln248Ala, and Tyr288Ala). Using these mutants, we performed whole-cell patch-clamp recording to analyze the effect of PPNDS on these mutants (*Figure 6B*).

In the mutational analysis, among three mutations to the lysine residue, two (Gln143Lys and Ile214Lys) showed significantly increased sensitivity to PPNDS (*Figure 6B*). In addition, the Gln248Lys

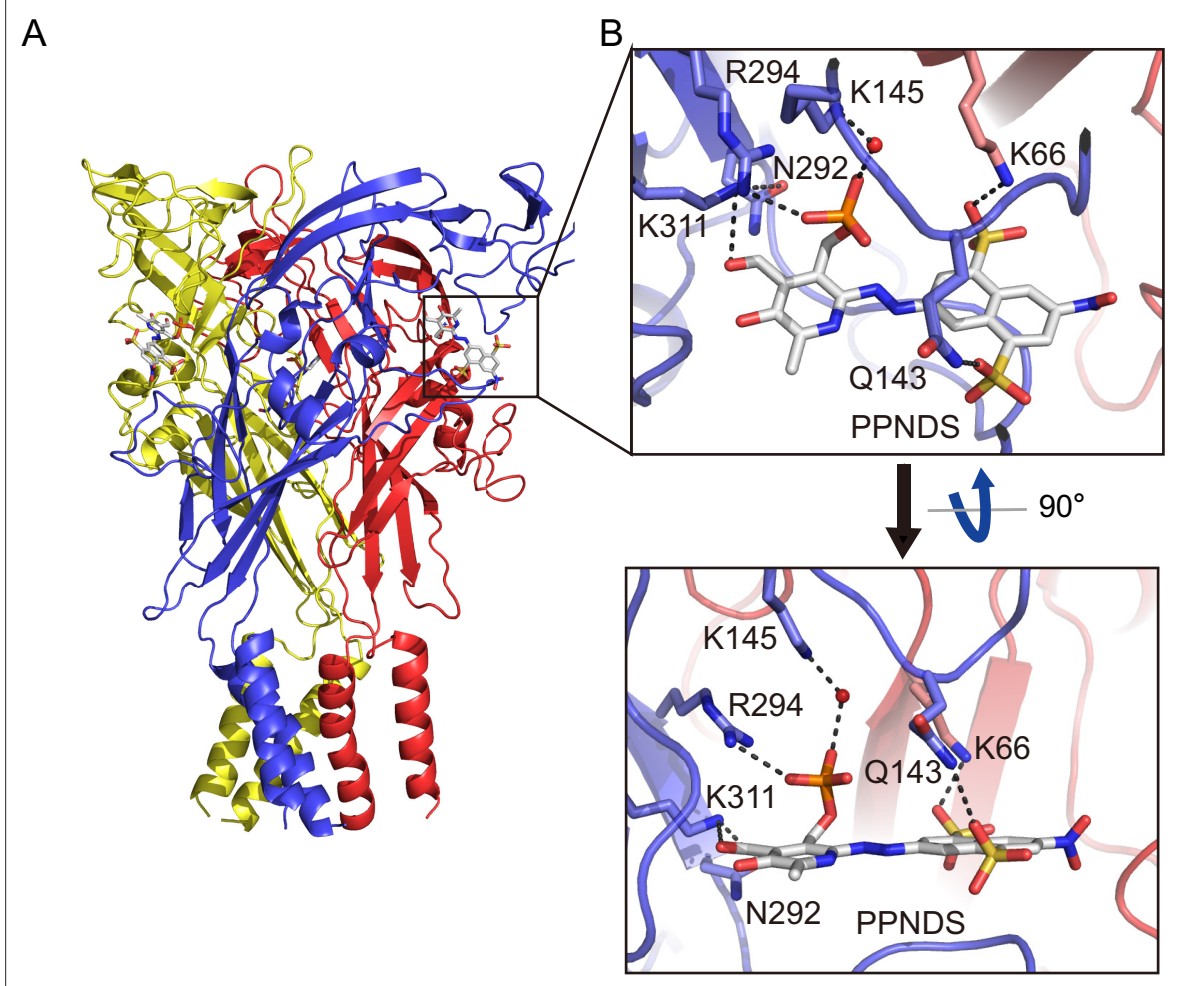

**Figure 2.** Binding site for pyridoxal-5'-phosphate-6-(2'-naphthylazo-6'-nitro-4',8'-disulfonate) (PPNDS). (**A, B**) Overall structure (**A**) and close-up view of the PPNDS binding site (**B**) in the PPNDS-bound panda P2X7 (pdP2X7) structure. PPNDS molecules are shown by stick models. Water molecules are depicted as red spheres. Dotted black lines indicate hydrogen bonding.

The online version of this article includes the following source data and figure supplement(s) for figure 2:

**Figure supplement 1.** EM density maps for the pyridoxal-5'-phosphate-6-(2'-naphthylazo-6'-nitro-4',8'-disulfonate) (PPNDS) and pyridoxal phosphate-6-azophenyl-2',5'-disulfonic acid (PPADS) binding sites.

**Figure supplement 2.** Molecular dynamics (MD) simulations of the pyridoxal-5'-phosphate-6-(2'-naphthylazo-6'-nitro-4',8'-disulfonate) (PPNDS)-bound panda P2X7 (pdP2X7) structure.

**Figure supplement 2—source data 1.** Numerical data for *Figure 2—figure supplement 2A, B*.

mutant showed slightly higher sensitivity to PPNDS, but not as significant an increase as the other two mutants (*Figure 6B*). Interestingly, while Tyr288 in pdP2X7 corresponds to Ser286 in hP2X1, the mutation of Tyr288 to alanine instead of serine significantly increased the affinity for PPNDS (*Figure 6B*). In addition, mutating Val173 in pdP2X7 to aspartate significantly reduced the sensitivity to PPNDS (*Figure 6B*). The two mutants Gln143Ala and Gln248Ala showed little or no decrease in PPNDS sensitivity (*Figure 6B*).

Following the result of the mutation analysis of pdP2X7, we then designed alanine mutants of hP2X1 at Lys140 (Gln143 in pdP2X7), Asp170 (Val173 in pdP2X7), Lys215 (Ile214 in pdP2X7), Lys249 (Gln248 in pdP2X7), and Ser286 (Tyr288 in pdP2X7) (*Figures 4B and 6A*). Using these mutants, we then performed whole-cell patch-clamp recording to evaluate the effect of PPNDS on these mutants (*Figure 6C*). All three mutants at the lysine residues (Lys140Ala, Lys215Ala, and Lys249Ala), especially the Lys140Ala and Lys215Ala mutants, showed a significant decrease in PPNDS sensitivity (*Figure 6C*). This result is largely consistent with the corresponding lysine-substituted mutants of

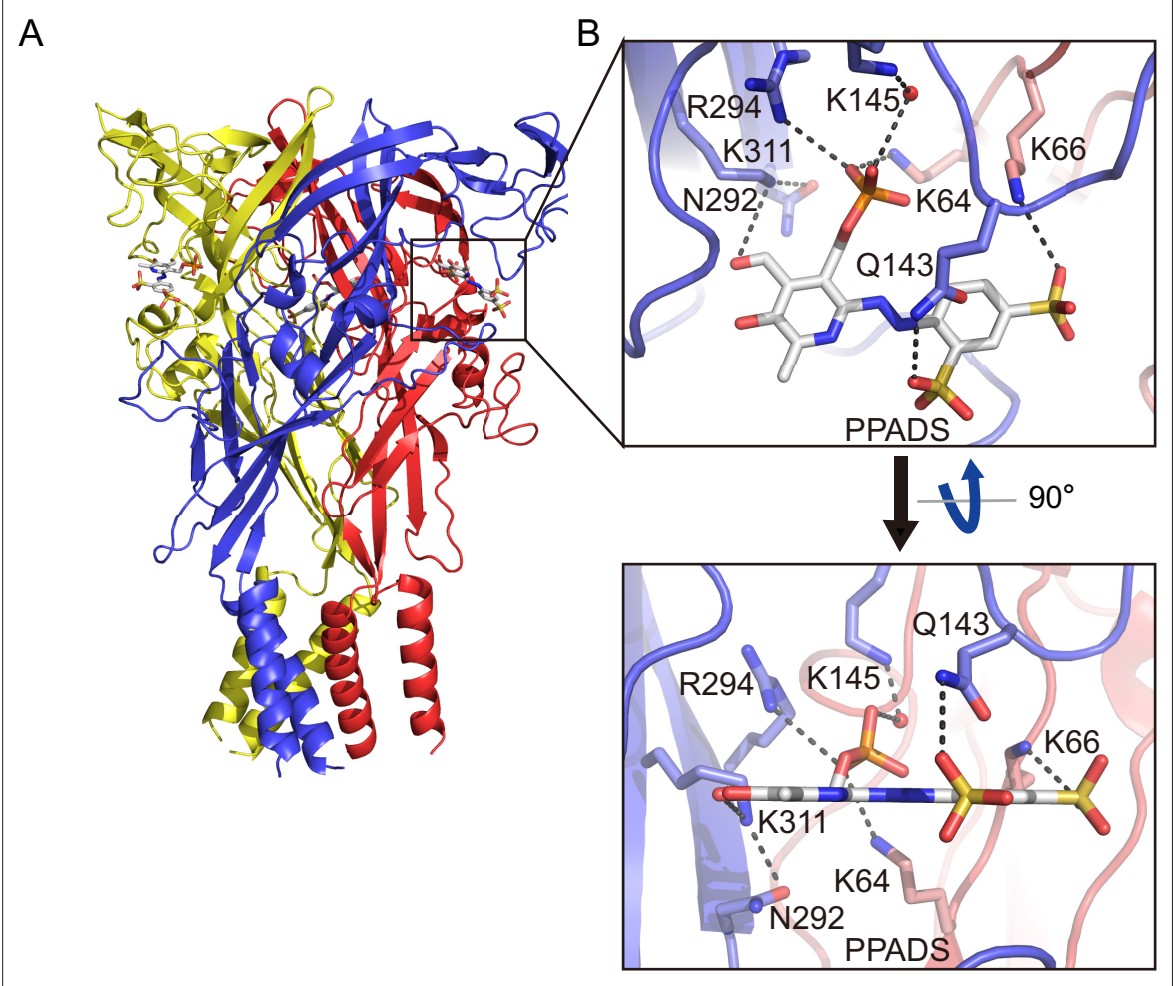

**Figure 3.** Binding site for pyridoxal phosphate-6-azophenyl-2′,5′-disulfonic acid (PPADS). (**A, B**) Overall structure (**A**) and close-up view of the PPADS binding site (**B**) in the PPADS-bound panda P2X7 (pdP2X7) structure. PPADS molecules are shown by stick models. Water molecules are depicted as red spheres. Dotted black lines indicate hydrogen bonding.

pdP2X7 (*Figure 6B*). Interestingly, the mutation of Asp170 and Ser286 to alanine increased the sensitivity to PPNDS (*Figure 6C*). Consistently, in the mutational analysis of the corresponding residues in pdP2X7, the mutation of Val173 to aspartate decreased PPNDS sensitivity, and the mutation of Tyr288 to alanine increased PPNDS sensitivity (*Figure 6B*).

Finally, among the three Lys residues involved in PPNDS sensitivity in hP2X1 (Lys140, Lys215, Lys249), Lys215 is also conserved in hP2X3 (Lys201). Accordingly, we performed mutational analysis of the hP2X3 Lys201Ala mutant (*Figure 6D*). As expected, the Lys201 mutant showed a decrease in PPNDS sensitivity (*Figure 6D*).

In summary, our mutational analysis based on structural comparison and sequence alignment identified several key residues involved in PPNDS sensitivity, particularly the residues involved in the subtype-specific difference in affinity for PPNDS.

## Discussion

In this work, we determined the cryo-EM structure of pdP2X7 in complex with two classical non-ATP analog inhibitors, PPNDS and PPADS, of pyridoxal phosphate derivatives (*Figure 1*) and revealed their orthosteric binding site (*Figures 2 and 3*). The binding site for PPNDS and PPADS has high overlap with the ATP-binding site (*Figure 4*). In the cryo-EM structures, the phosphate group of PPNDS and PPADS appears to occupy the position of the γ-phosphate group of ATP in the ATP-bound structure (*Figures 2–4*). Comparison with the previously reported apo- (closed) and ATP-bound (open)

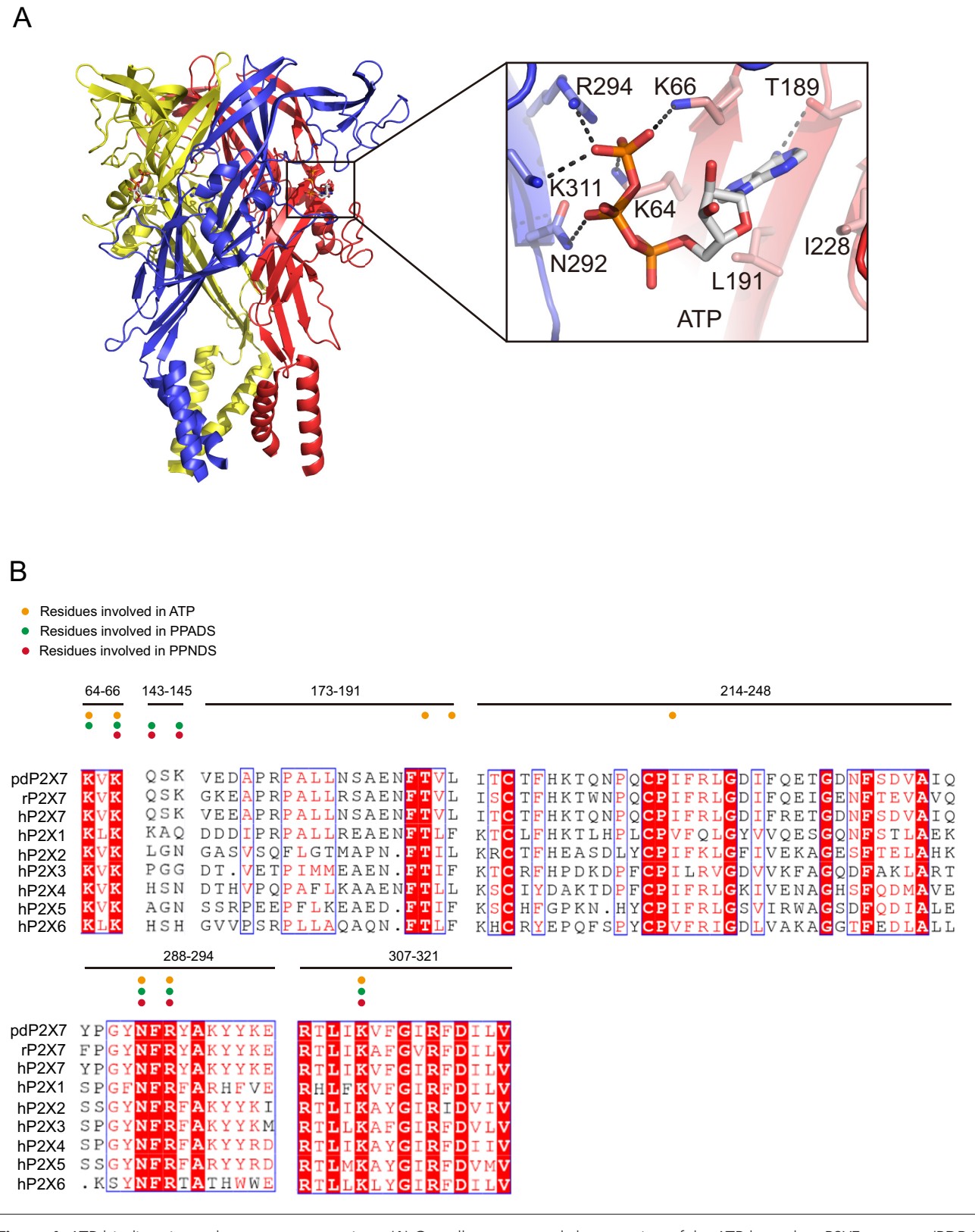

**Figure 4.** ATP-binding site and sequence comparison. (**A**) Overall structure and close-up view of the ATP-bound rat P2X7 structure (PDB ID: 6U9W). The cytoplasmic domain is not shown. Dotted black lines indicate hydrogen bonding. (**B**) Sequence alignment of *A. melanoleuca* P2X7 (pdP2X7) (accession number: XP_002913164.3), *Rattus norvegicus* (rP2X7) (accession number: Q64663.1), and *Homo sapiens* P2X receptors (P2X1: P51575.1, P2X2: Q9UBL9.1, P2X3: P56373.2, P2X4: Q99571.2, P2X5: Q93086.4, P2X6: O15547.2, and P2X7: Q99572.4). Orange, green, and red circles indicate the residues involved in ATP, PPADS, and PPNDS recognition.

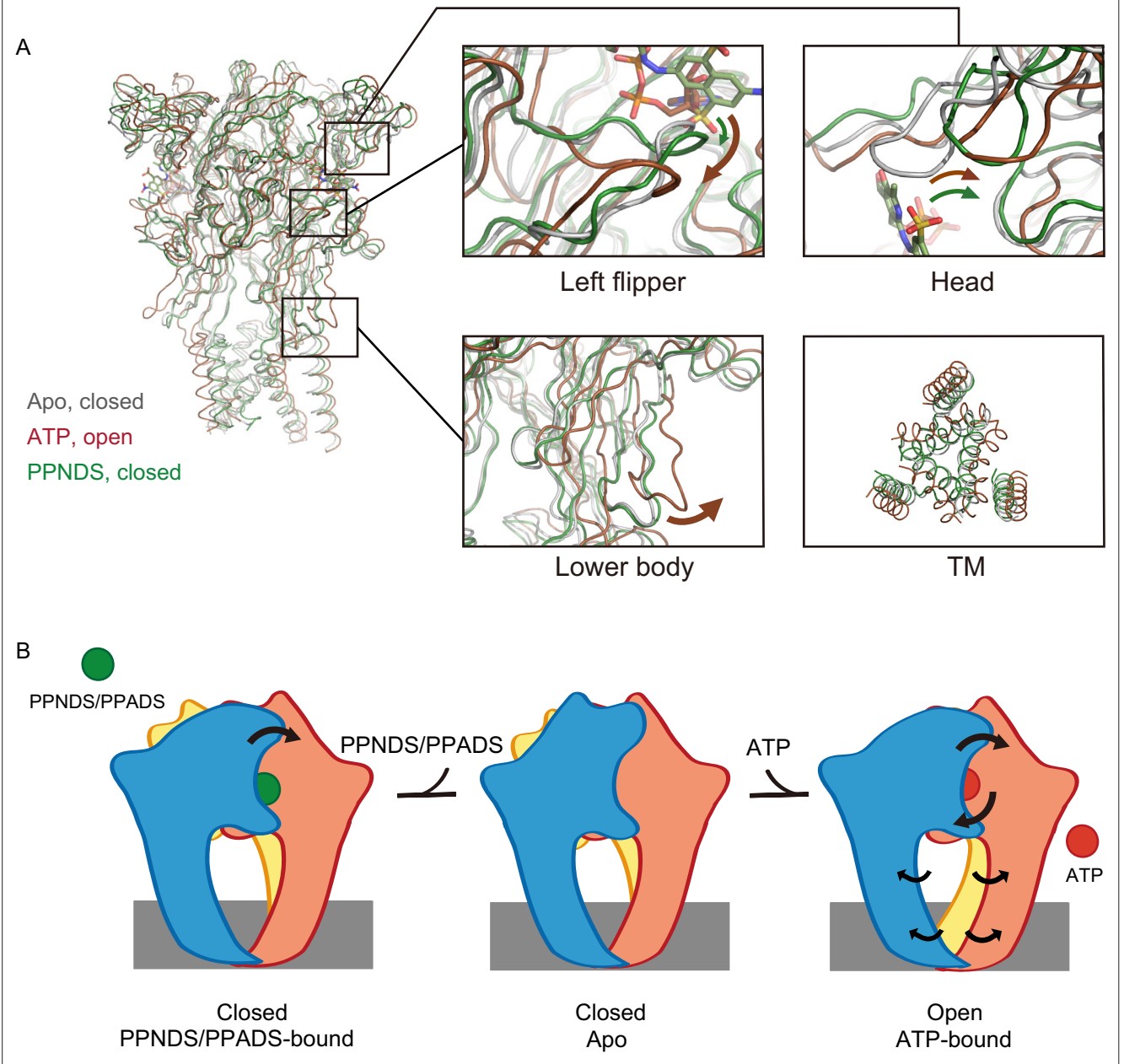

**Figure 5.** Structural comparison and inhibition mechanism. (**A**) Superposition of the ATP-bound rP2X7 structure (red, PDB ID: 6U9W) and the pyridoxal-5'-phosphate-6-(2'-naphthylazo-6'-nitro-4',8'-disulfonate) (PPNDS)-bound panda P2X7 (pdP2X7) structure (green, this study) onto the apo rP2X7 structure (gray, PDB ID: 6U9V). Close-up views of the head, left flipper, and lower body domains and the intracellular view of the transmembrane (TM) domain are shown in each box. Arrows indicate the conformational changes from the apo- to ATP-bound states (red) and from the apo- to the PPNDS-bound states (green). (**B**) A cartoon model of the PPNDS/pyridoxal phosphate-6-azophenyl-2',5'-disulfonic acid (PPADS)-dependent inhibition and ATP-dependent activation mechanisms.

structures showed that, in contrast to ATP binding, the binding of PPNDS and PPADS does not induce the downward motion of the left flipper, which is essential for channel activation (*Figure 5*). These observations provide mechanistic insights into channel inhibition by pyridoxal phosphate derivatives. Finally, structure-based mutational analyses revealed several key residues important for PPNDS sensitivity, particularly for the subtype-specific difference in sensitivity (*Figure 6*).

Besides PPADS and PPNDS in this study, TNP-ATP is well known as a classical competitive inhibitor for P2X receptors (*North and Surprenant, 2000*). In the previously reported TNP-ATP-bound P2X7 structure, TNP-ATP binding induces the conformational changes of the left flipper region but not of

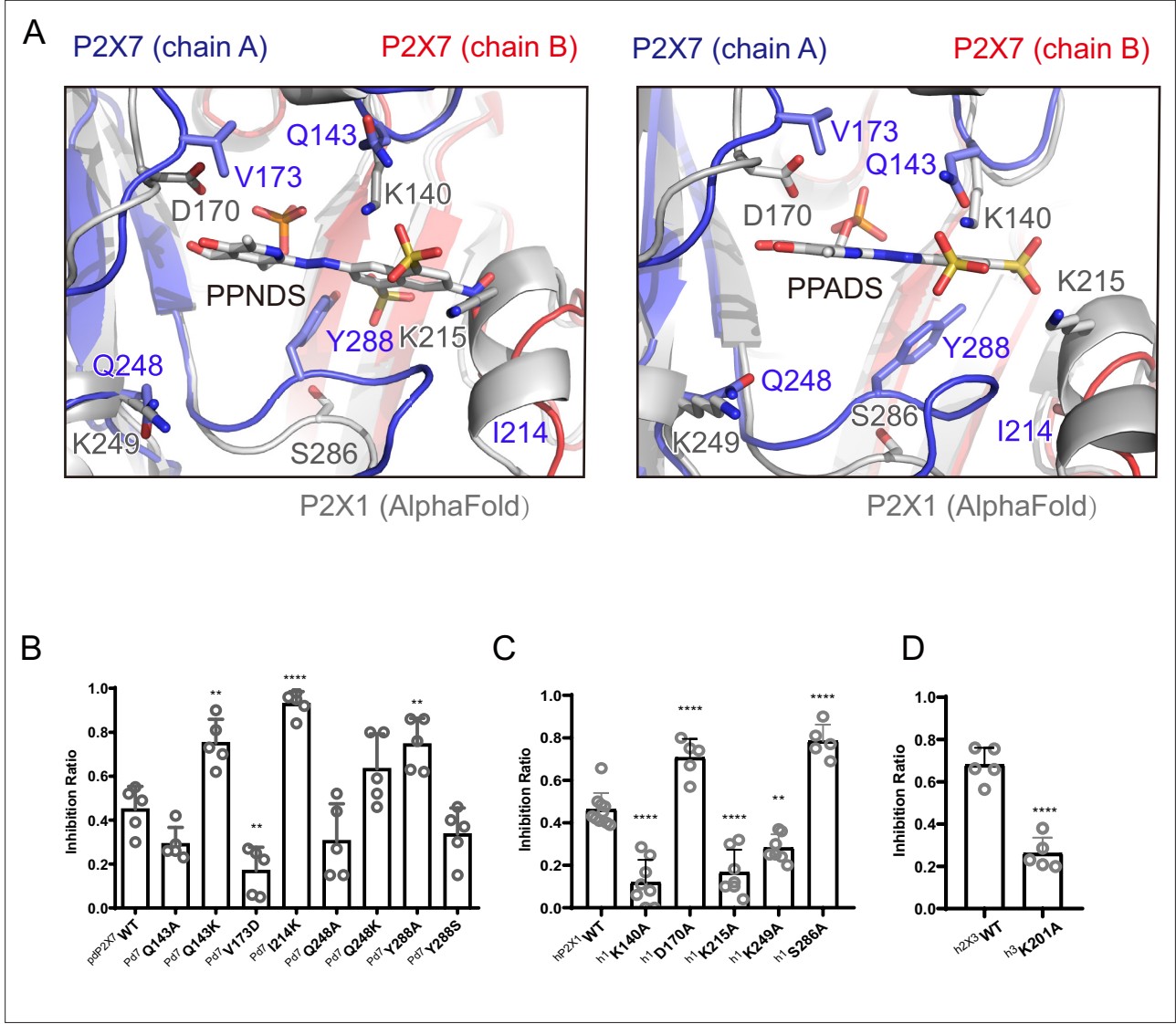

**Figure 6.** Structure-based mutational analysis. (**A**) Superimposition of the pyridoxal-5'-phosphate-6-(2'-naphthylazo-6'-nitro-4',8'-disulfonate) (PPNDS)-bound and pyridoxal phosphate-6-azophenyl-2',5'-disulfonic acid (PPADS)-bound structures in this study onto the predicted human P2X1 (hP2X1) structure (AlphaFold). Each subunit of the PPNDS-bound and PPADS-bound structures is shown in blue, yellow, and red, while the predicted hP2X1 structure is shown in gray. The PPNDS and PPADS molecules and the residues surrounding PPNDS and PPADS that are different between panda P2X7 (pdP2X7) and hP2X1 are shown as sticks. (**B**) Effects of PPNDS (10 µM) on ATP (1 mM)-evoked currents of pdP2X7 and its mutants (mean ± SD, n=5, one-way ANOVA post hoc test, **: p<0.01, ****: p<0.0001 vs. WT). (**C**) Effects of PPNDS (1 µM) on ATP (1 µM)-evoked currents of hP2X1 and its mutants (mean ± SD, n=5–10, one-way ANOVA post hoc test, **: p<0.01, ****: p<0.0001 vs. WT). (**D**) Effects of PPNDS (10 µM) on ATP (1 µM)-evoked currents of hP2X3 and its mutants (mean ± SD, n=5, unpaired t test, ****: p<0.0001 vs. WT).

The online version of this article includes the following source data and figure supplement(s) for figure 6:

**Source data 1.** Numerical data for *Figure 6B–D*.

**Figure supplement 1.** Effects of pyridoxal-5'-phosphate-6-(2'-naphthylazo-6'-nitro-4',8'-disulfonate) (PPNDS) on P2X receptors by patch-clamp recording.

the head domain (*Kasuya et al., 2017*). In contrast, PPADS and PPNDS binding induces the conformational changes of the head domain but not of the left flipper region (*Figure 5*). These contrasts would highlight the uniqueness of the competitive inhibition mechanism by pyridoxal phosphate derivatives as well as the diversity of competitive inhibition mechanisms of P2X receptors.

There are several previous reports characterizing the P2X receptor binding site for pyridoxal phosphate derivatives, especially PPADS (*Huo et al., 2018*; *Michel et al., 2008*; *Buell et al., 1996*;

*Garcia-Guzman et al., 1997*). The mutation of Glu249 in rat P2X4 (rP2X4) to lysine was designed based on the corresponding residue in hP2X1 (Lys249) to confer PPADS sensitivity on the PPADS-insensitive P2X4 receptor (*Buell et al., 1996*). Glu249 in rP2X4 corresponds to Gln248 in pdP2X7 (*Figure 4B*). In our structures, this residue is proximal to the hydroxyl moieties of the pyridoxal phosphate group of PPADS and PPNDS (*Figure 6A*). More recently, the combination of docking simulation and electrophysiology showed that Lys70, Asp170, Lys190, and Lys249 participate in PPADS binding in hP2X1 (*Huo et al., 2018*). Among these four residues, Lys70 and Lys190 are directly involved in ATP binding (*Hattori and Gouaux, 2012*), and Lys249 corresponds to Glu249 in rP2X4, as shown in a previous study (*Buell et al., 1996*). In addition, the mutation of Asp170 to cysteine increased sensitivity to PPADS (*Huo et al., 2018*). Consistently, our mutational analysis revealed that the Asp170Ala mutation in hP2X1 also increased the sensitivity to PPNDS (*Figure 6C*). According to our structures, the corresponding residue in pdP2X7 (Val173) is located proximal to the pyridoxal phosphate group of PPADS, and the pyridoxal phosphate group is in common with that of PPADS (*Figure 6A*). Therefore, Asp170 in hP2X1 is involved in sensitivity to both PPADS and PPNDS.

It should be noted that the orientation of PPADS in the recent docking model is very different from that in our cryo-EM structure. In the docking model, compared to our cryo-EM structure, PPADS shows an almost 180° rotation relative to the axis of rotation parallel to the membrane with the azide group of PPADS as the fulcrum (*Huo et al., 2018*). This difference in the orientation of the compound may explain why hP2X1 Lys140 and Lys215 were not predicted to be involved in PPADS binding in the previous docking model, in contrast to the results of our study.

In the past, several types of pyridoxal phosphate derivatives have been identified as P2X inhibitors (*Jacobson et al., 2002*; *Cho et al., 2013*). However, most of them share a common weakness in subtype specificity, where they tend to show high affinity for both P2X1 and P2X3. For example, MRS 2257 was identified as the most active PPADS analog from the screening but has IC50 values of 5 nM and 22 nM for P2X1 and P2X3, respectively (*Brown et al., 2001*). In addition, another PPADS analog, termed 36j, shows improved subtype specificity for P2X3 (IC50: 60 nM for P2X3) but still shows a moderate inhibitory effect on P2X1 at 10 μM (*Cho et al., 2013*). Thus, it is still difficult to obtain competitive inhibitors of P2X receptors, including pyridoxal phosphate derivatives, with subtype specificity. This is probably because the residues directly involved in ATP binding are strictly conserved among the subtypes (*Figure 4B*). To overcome this situation, our work might facilitate the rational design of pyridoxal phosphate derivatives with strict subtype specificity for P2X receptors. We have not only defined the binding mode of pyridoxal phosphate derivatives to P2X receptors but also newly identified a subtype-specific residue for pyridoxal phosphate derivatives. For example, Lys215 and Lys249 in hP2X1 are important for PPNDS sensitivity (*Figure 6C*) but are not conserved in other P2X subtypes, including P2X3 (*Figure 4B*). These findings would provide a clue for the design of more subtype-specific pyridoxal phosphate derivatives.

In conclusion, our structural and functional analyses provided mechanistic insights into the orthosteric inhibition mechanism of P2X receptors by the classical pyridoxal phosphate derivative P2X antagonist. In addition, we identified key residues involved in compound sensitivity, especially the differential sensitivity between P2X subtypes. These results potentially facilitate the development of subtype-specific compounds targeting P2X receptors, which have attracted widespread interest as therapeutic targets.

## Methods

### Expression and purification of P2X7

The previously reported functional expression construct of giant panda (*A. melanoleuca*) P2X7 for structural studies (pdP2X7, residues 22–359, N241S/N284S/V35A/R125A/E174K, XP_002913164.1) (*Karasawa and Kawate, 2016*) was synthesized (Genewiz, China), subcloned, and inserted into a modified version of the pFastBac vector (Invitrogen, USA) with an octahistidine tag, Twin-Strep-tag, mEGFP, and a human rhinovirus (HRV) 3C protease cleavage site at the N-terminus. Using the Bac-to-Bac system, the mEGFP-fusion pdP2X7 construct was expressed in Sf9 cells infected with baculovirus. The Sf9 cells were collected by centrifugation (5400×*g*, 10 min) and subsequently disrupted using an ultrasonic homogenizer in TBS buffer (20 mM Tris pH 8.0, 150 mM NaCl) containing 1 mM phenylmethylsulfonyl fluoride, 5.2 μg/mL aprotinin, 1.4 μg/mL pepstatin, and 1.4 μg/mL leupeptin.

The supernatant was harvested after centrifugation (7600×$g$, 20 min). The membrane fraction was then isolated by ultracentrifugation (200,000×$g$, 1 hr) and solubilized in buffer A (50 mM Tris pH 7.5, 150 mM NaCl) containing 2% (wt/vol) $n$-dodecyl-beta-D-maltopyranoside (DDM) at 4°C for 1 hr. The solubilized supernatant was collected by another round of ultracentrifugation (200,000×$g$, 1 hr) and applied to a Strep-Tactin resin column (QIAGEN, USA) equilibrated with buffer A containing 0.025% (wt/vol) DDM. The resin was incubated for 1 hr, and the column was eluted with buffer B (100 mM Tris pH 8.0, 150 mM NaCl, 2.5 mM desthiobiotin, 0.025% [wt/vol] DDM). The eluted protein was concentrated to 1 mg/mL before being prepared for nanodisc reconstitution.

### Nanodisc reconstitution

Soybean polar lipid extract (Avanti Polar Lipids, USA) was dissolved in chloroform, dried under a nitrogen stream, and then resuspended in reconstitution buffer (20 mg/mL soybean polar lipid, 20 mM HEPES pH 7.0, and 150 mM NaCl). Following a 1 hr incubation at room temperature, the lipid suspension was subjected to sonication for 5 min until the lipids reached a near-transparent state. Subsequently, DDM (Anatrace, USA) was added at a final concentration of 0.4% and incubated at room temperature overnight. The mEGFP-fusion pdP2X7, MSP2N2 protein, and soybean polar lipid were combined in a molar ratio of 1:3:180. This mixture was then incubated at 4°C for 1 hr and further subjected to a 4 hr incubation with bio-beads (Bio-Rad, USA). After incubation, the bio-beads were removed via filtration, and the nanodisc fractions containing mEGFP-fusion pdP2X7 were bound to Ni-NTA (QIAGEN, USA) resin preequilibrated with wash buffer (20 mM HEPES pH 7.5, 150 mM NaCl, 30 mM imidazole) and subsequently eluted using elution buffer (20 mM HEPES pH 7.5, 150 mM NaCl, 300 mM imidazole). To cleave the N-terminal EGFP, the elution was mixed with HRV3C protease and incubated at room temperature for 1 hr, followed by overnight incubation at 4°C. The nanodisc-reconstituted pdP2X7 protein was separated through size-exclusion chromatography using a Superdex 200 Increase 10/300 column (Cytiva, USA) preequilibrated with SEC buffer (20 mM HEPES pH 7.5 and 150 mM NaCl) and subsequently concentrated to 0.9 mg/mL. P2X7 antagonists (PPNDS or PPADS) were added to the nanodisc-reconstituted pdP2X7 at a final concentration of 50 µM and incubated on ice for 1 hr before cryo-EM grid preparation.

### EM data acquisition

For both the PPNDS-bound and PPADS-bound pdP2X7 samples, a total of 2.5 µL of the nanodisc-reconstituted pdP2X7 was applied to a glow-discharged holey carbon-film grid (Quantifoil, Au 1.2/1.3, 300 mesh, USA). The grid was then blotted using a Vitrobot (Thermo Fisher Scientific, USA) system with a 3.0 s blotting time at 100% humidity and 4°C, followed by plunge-freezing in liquid ethane. Cryo-EM data collection was carried out using a 300 kV Titan Krios microscope (Thermo Fisher Scientific, USA) equipped with a K3 direct electron detector (Gatan Inc, USA). The specimen stage temperature was maintained at 80 K. Movies were recorded using beam-image shift data collection methods (*Wu et al., 2019*) in superresolution mode, with a pixel size of 0.41 Å (physical pixel size of 0.83 Å), a magnification of ×29,000, and defocus values ranging from –1.3 µm to –2.0 µm. The dose rate was set to 20 e–/s, and each movie consisted of 40 frames with an exposure of 50 e–/Å, resulting in each movie being 1.724 s long.

### Image processing

A total of 9664 and 4692 movies for the PPNDS-bound and PPADS-bound pdP2X7 samples, respectively, were motion-corrected and binned with MotionCor2 (*Zheng et al., 2017*) with a patch of 5×5, producing summed and dose-weighted micrographs with a pixel size of 0.83 Å. Contrast transfer function parameters were estimated by CTFFIND 4.1 (*Rohou and Grigorieff, 2015*). Particle picking and 2D classification were performed using RELION 3.1 (*Zivanov et al., 2018*). In total, 1,537,753 particles for the PPNDS-bound sample and 1,835,907 particles for the PPADS-bound sample were autopicked and extracted using a box size of 256×256 pixels. After 2D classification, we performed 3D classification with C1 symmetry using RELION 3.1 on 633,674 particles for the PPNDS-bound sample and 236,753 particles for the PPADS-bound sample. Then, 121,008 particles for the PPNDS-bound sample and 161,188 particles for the PPADS-bound sample were selected for non-uniform refinement using cryoSPARCv4.2.1 (*Punjani et al., 2020*), applying C3 symmetry for the final 3D reconstruction. The resulting resolutions of the PPNDS-bound and PPADS-bound pdP2X7 structures were 3.3 Å and 3.6 Å,

respectively, as determined by the Fourier shell correlation (FSC)=0.143 criterion on the corrected FSC curves. The local resolution was estimated using cryoSPARCv4.2.1. The workflows for image processing and for 3D reconstruction are shown in *Figure 1—figure supplements 2–5*. The figures were generated by UCSF Chimera (*Pettersen et al., 2004*).

## Model building

The initial models of pdP2X7 were manually built starting from the previously reported pdP2X7 structure (PDB ID: 5U1L). Manual model building was performed using Coot (*Emsley et al., 2010*). Real-space refinement was performed using PHENIX (*Liebschner et al., 2019*). All structure figures were generated using PyMOL (https://pymol.org/). For the predicted structure of hP2X1, the previously generated model using AlphaFold and ColabFold was used (*Jumper et al., 2021*; *Mirdita et al., 2022*; *Shen et al., 2023*). The sequence alignment figure was generated using Clustal Omega (*Sievers and Higgins, 2018*) and ESPript 3.0 (*Robert and Gouet, 2014*).

## Cell lines

HEK293 cells were purchased from Shanghai Institutes for Biological Sciences and cultured in Dulbecco's modified Eagle's medium supplemented with 10% fetal bovine serum, 1% penicillin-streptomycin, and 1% GlutaMAX at 37°C in a humidified atmosphere of 5% $CO_2$ and 95% air (*Wang et al., 2017*; *Sun et al., 2022*). This cell line tested negative for mycoplasma contamination and has been done by the vendor.

## Electrophysiology

Plasmids harboring hP2X1, hP2X3, or pdP2X7 were transfected into cells by calcium phosphate transfection (*Zhang et al., 2022*). Currents of hP2X1 and hP2X3 were recorded using nystatin (Sangon Biotech, China) perforated recordings to prevent rundown in current during multiple dose applications of ATP. Nystatin (0.15 mg/mL) was diluted with a high-potassium internal intracellular solution containing 75 mM $K_2SO_4$, 55 mM KCl, 5 mM $MgSO_4$, and 10 mM HEPES (pH 7.4). Currents of PdP2X7 receptors were recorded using a conventional whole-cell patch configuration. After 24–48 hr of transfection, HEK293 cells were recorded at room temperature (25 ± 2°C) using an Axopatch 200B amplifier (Molecular Devices, USA) with a holding potential of –60 mV. Current data were sampled at 10 kHz, filtered at 2 kHz, and analyzed using pCLAMP 10 (Molecular Devices, USA) for analysis. HEK293 cells were bathed in standard extracellular solution (SS) containing 2 mM $CaCl_2$, 1 mM $MgCl_2$, 10 mM HEPES, 150 mM NaCl, 5 mM KCl, and 10 mM glucose with the pH adjusted to 7.4. For conventional whole-cell recordings, the pipette solutions consisted of 120 mM KCl, 30 mM NaCl, 0.5 mM $CaCl_2$, 1 mM $MgCl_2$, 10 mM HEPES, and 5 mM EGTA with pH adjusted to 7.4. ATP and other compounds were dissolved in SS for P2X1 and P2X3 and applied to Y-tubes. For pdP2X7, ATP and other compounds were dissolved in 0 Ca, 0 Mg solution containing 150 mM NaCl, 10 mM glucose, 10 mM HEPES, 5 mM KCl, and 10 mM EGTA with the pH adjusted to 7.4 (*Cui et al., 2022*). PPNDS was purchased from APE×BIO, and PPADS was purchased from MCE. The standard solution and 0 Ca, 0 Mg solution were formulated with compounds from Aladdin, and internal solutions were formulated with compounds from Sigma-Aldrich (*Li et al., 2018*). All electrophysiological recordings were analyzed using Clampfit 10.6 (Molecular Devices, USA). Pooled data are expressed as the mean and standard error (s.e.m.). Statistical comparisons were made using Bonferroni's post hoc test (ANOVA). **p<0.01 and ***p<0.0001 were considered significant.

## MD simulations

The energy-minimized models of the PPNDS-bound pdP2X7 were used as the initial structures for MD simulations. A large 1-palmitoyl-2-oleoyl-*sn*-glycero-3-phosphocholine (300 K) bilayer, available in System Builder of DESMOND (*Zhang et al., 2022*; *Dror et al., 2011*), was built to generate a suitable membrane system based on the OPM database (*Lomize et al., 2012*). The systems were dissolved in simple point charge (SPC) water molecules. The DESMOND default relaxation protocol was applied to each system prior to the simulation run. (1) 100 ps simulations in the NVT (constant number [N], volume [V], and temperature [T]) ensemble with Brownian kinetics using a temperature of 10 K with solute heavy atoms constrained; (2) 12 ps simulations in the NVT ensemble using a Berendsen thermostat with a temperature of 10 K and small-time steps with solute heavy atoms constrained; (3) 12

ps simulations in the NPT (constant number [N], pressure [P], and temperature [T]) ensemble using a Berendsen thermostat and barostat for 12 ps simulations at 10 K and 1 atm, with solute heavy atoms constrained; (4) 12 ps simulations in the NPT ensemble using a Berendsen thermostat and barostat at 300 K and 1 atm with solute heavy atoms constrained; and (5) 24 ps simulations in the NPT ensemble using a Berendsen thermostat and barostat at 300 K and 1 atm without constraint. After equilibration, the MD simulations were performed for 0.3 μs. The long-range electrostatic interactions were calculated using the smooth particle grid Ewald method. The trajectory recording interval was set to 200 ps, and the other default parameters of DESMOND were used in the MD simulation runs. All simulations used the all-atom OPLS_2005 force field (*Jorgensen et al., 1996*; *Kaminski et al., 2001*; *Banks et al., 2005*), which is used for proteins, ions, lipids and SPC waters. The Simulation Interaction Diagram (SID) module in DESMOND was used for exploring the interaction analysis between PPNDS and pdP2X7. All simulations were performed on a DELL T7920 with NVIDIA TESLTA K40C or CAOWEI 4028GR with NVIDIA TESLTA K80. The simulation system was prepared, and the trajectory was analyzed and visualized on a CORE DELL T7500 graphics workstation with 12 CPUs.

## Statistics

Electrophysiological recordings were repeated 5–10 times. Error bars represent the standard error of the mean. Cryo-EM data collection and refinement statistics are summarized in *Table 1*.

## Acknowledgements

We thank the staff scientists at the Center for Biological Imaging, Institute of Biophysics, Chinese Academy of Sciences for technical assistance with cryo-EM data collection (project numbers: CBIapp202007004; 2020-NFPS-PT-005280). This work was supported by funding from the National Natural Science Foundation of China to MH (32071234, 32271244, and 32250610205). This work was also supported by the Innovative Research Team of High-level Local Universities in Shanghai, a key laboratory program of the Education Commission of Shanghai Municipality (ZDSYS14005). This work was also supported by JST, PRESTO Grant Number JPMJPR20E1, Japan to MI.

## Additional information

### Funding

| Funder | Grant reference number | Author |
|---|---|---|
| National Natural Science Foundation of China | 32071234 | Motoyuki Hattori |
| National Natural Science Foundation of China | 32271244 | Motoyuki Hattori |
| National Natural Science Foundation of China | 32250610205 | Motoyuki Hattori |
| Shanghai Municipal Education Commission | ZDSYS14005 | Motoyuki Hattori |
| Japan Science and Technology Agency | 10.52926/JPMJPR20E1 | Muneyoshi Ichikawa |

The funders had no role in study design, data collection and interpretation, or the decision to submit the work for publication.

### Author contributions

Danqi Sheng, Chen-Xi Yue, Data curation, Formal analysis, Validation, Investigation, Visualization, Writing – original draft; Fei Jin, Yao Wang, Data curation; Muneyoshi Ichikawa, Formal analysis, Funding acquisition; Ye Yu, Resources, Formal analysis, Validation, Investigation; Chang-Run Guo, Conceptualization, Formal analysis, Supervision, Validation, Investigation, Visualization, Writing – original draft, Project administration; Motoyuki Hattori, Conceptualization, Resources, Formal analysis,

Supervision, Funding acquisition, Validation, Investigation, Visualization, Writing – original draft, Project administration

### Author ORCIDs
Muneyoshi Ichikawa http://orcid.org/0000-0002-5921-7699
Chang-Run Guo http://orcid.org/0009-0004-2348-3591
Motoyuki Hattori https://orcid.org/0000-0002-5327-5337

Reviewer #1 (Public Review): https://doi.org/10.7554/eLife.92829.3.sa1
Reviewer #2 (Public Review): https://doi.org/10.7554/eLife.92829.3.sa2
Author response https://doi.org/10.7554/eLife.92829.3.sa3

## Additional files

### Supplementary files
• MDAR checklist

### Data availability
The atomic coordinates and structural factors for the pdP2X7 in complex with PPNDS (PDB: 8JV8 and EMD-36671) and PPADS (PDB: 8JV7 and EMD-36670) have been deposited in the Protein Data Bank. All other relevant data are included in the paper or its figure supplements, including the source data files, or deposited in ScienceDB (https://doi.org/10.57760/sciencedb.11168).

The following datasets were generated:

| Author(s) | Year | Dataset title | Dataset URL | Database and Identifier |
| --- | --- | --- | --- | --- |
| Sheng D, Hattori M | 2023 | Cryo-EM structures of the panda P2X7 receptor in complex with PPNDS | https://www.rcsb.org/structure/8JV8 | RCSB Protein Data Bank, 8JV8 |
| Sheng D, Hattori M | 2023 | Cryo-EM structures of the panda P2X7 receptor in complex with PPADS | https://www.rcsb.org/structure/8JV7 | RCSB Protein Data Bank, 8JV7 |
| Sheng D, Hattori M | 2023 | Cryo-EM structures of the panda P2X7 receptor in complex with PPNDS | https://www.ebi.ac.uk/emdb/EMD-36671 | EMDB, EMD-36671 |
| Sheng D, Hattori M | 2023 | Cryo-EM structures of the panda P2X7 receptor in complex with PPADS | https://www.ebi.ac.uk/emdb/EMD-36670 | EMDB, EMD-36670 |
| Sheng D, Yue CX, Jin F, Wang Y, Ichikawa M, Yu Y, Guo CR, Hattori M | 2023 | Electrophysiology of P2X receptors and their mutants | https://doi.org/10.57760/sciencedb.11168 | Science Data Bank, 10.57760/sciencedb.11168 |

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
