## [Editor Report · eLife assessment]

This study by Sheng and colleagues provides **valuable** insights into the mechanism of competitive inhibitors of P2X receptors. The structural and functional evidence supporting the subtype specificity of pyridoxal-5'-phosphate derivatives is **compelling** and provides information for designing drugs that selectively target different subtypes of P2X receptor channels. The work will be of interest to biochemists, structural biologists, and pharmacologists.

---

## [Referee Report · Reviewer #1 (Public Review)]

This work provides new mechanistic insights into the competitive inhibition in the mammalian P2X7 receptors using structural and functional approaches. The authors solved the structure of panda (pd) P2X7 in the presence of the classical competitive antagonists PPNDS and PPADS. They find that both the drugs bind to the orthosteric site employed by the physiological agonist ATP. However, owing to the presence of a single phosphate group, they prevent movements in the flipper domain required for channel opening. The authors performed structure based mutational analysis together with electrophysiological characterization to understand the subtype specific binding of these drugs. It is known from previous studies that P2X1 and P2X3 are more sensitive to these drugs as compared to P2X7, hence, the residues adjacent to the ATP binding site in pdP2X7 were mutated to those present in P2X1. They observed that mutations of Q143, I214 and Q248 into lysine (hP2X1) increased the P2X7 sensitivity to PPNDS, whereas in P2X1, mutations of these lysines to alanine reduced sensitivity to PPNDS, suggesting that these key residues contribute to the subunit specific sensitivity to these drugs. Similar experiments were done in hP2X3 to demonstrate its higher sensitivity to PPNDS. This preprint provides a useful framework for developing subtype specific drugs for the family of P2X receptor channels, an area that is currently relatively unexplored.

The conclusions of the paper are well supported.

The revised manuscript is well written and presents its data with more clarity.

---

## [Referee Report · Reviewer #2 (Public Review)]

Summary:

P2X receptors play pivotal roles in physiological processes such as neurotransmission and inflammation, making them promising drug targets. This study, through cryo-EM and functional experiments, reveals the structural basis of the competitive inhibition of the PPNDS and PPADS on mammalian P2X7 receptors. Key findings include the identification of the orthosteric site for these antagonists, the revelation of how PPADS/PPNDS binding impedes channel-activating conformational changes, and the pinpointing of specific residues in P2X1 and P2X3 subtypes that determine their heightened sensitivity to these antagonists. These insights present a comprehensive understanding that could guide the development of improved drugs targeting P2X receptors. This work will be a valuable addition to the field.

Strengths:

The combination of structural experiments and mutagenesis analyses offers a deeper understanding of the mechanism. While the inclusion of MD simulation is appreciated, providing more insights from the simulation might further strengthen this already compelling story.

---

## [Author Response]

The following is the authors’ response to the original reviews.

**Public Reviews:**

**Reviewer #1 (Public Review):**
This work provides new mechanistic insights into the competitive inhibition in the mammalian P2X7 receptors using structural and functional approaches. The authors solved the structure of panda (pd) P2X7 in the presence of the classical competitive antagonists PPNDS and PPADS. They find that both drugs bind to the orthosteric site employed by the physiological agonist ATP. However, owing to the presence of a single phosphate group, they prevent movements in the flipper domain required for channel opening. The authors performed structure-based mutational analysis together with electrophysiological characterization to understand the subtype-specific binding of these drugs. It is known from previous studies that P2X1 and P2X3 are more sensitive to these drugs as compared to P2X7, hence, the residues adjacent to the ATP binding site in pdP2X7 were mutated to those present in P2X1. They observed that mutations of Q143, I214, and Q248 into lysine (hP2X1) increased the P2X7 sensitivity to PPNDS, whereas in P2X1, mutations of these lysines to alanine reduced sensitivity to PPNDS, suggesting that these key residues contribute to the subunit-specific sensitivity to these drugs. Similar experiments were done in hP2X3 to demonstrate its higher sensitivity to PPNDS. This preprint provides a useful framework for developing subtype-specific drugs for the family of P2X receptor channels, an area that is currently relatively unexplored.

We appreciate the time and effort Reviewer #1 devoted to this review, and we have addressed the specific comments below.

(1) Why was the crystallization construct of panda P2X7 used for structural studies instead of rat P2X7 with the cytoplasmic ballast which is a more complete receptor that is closely related to the human receptor? Can the authors provide a justification for this choice?

We appreciate this comment. We did try to express the rat P2X7 receptor in its full-length form based on a previous report (Cell 2019, PMID: 31587896), but the expression of the receptor was not successful for an unknown reason. Instead, we employed a truncated construct of panda P2X7 based on the findings described another previous report (eLife 2016, PMID: 27935479). This truncated construct also possesses ATP-dependent channel activity (eLife 2016, PMID: 27935479). Thus, we understand that the full-length P2X7 construct would be preferable, particularly for addressing the function of the cytoplasmic domain; however, the main focus of this study was on PPNDS/PPNADS recognition and the associated structural changes in the ATP binding pocket, which we believe are less likely to be severely affected by truncation of the cytoplasmic domain. In support of this expectation, our mutational analyses are consistent with the structures in this study.Therefore, we believe that the use of the truncation construct in this study is justified.

(2) Was there a good reason why hP2X1 and hP2X3 currents were recorded in perforated patches, whereas pdP2X7 currents were recorded using the whole-cell configuration? It seems that the extent of rundown is less of a problem with perforated patch recordings. Can the authors comment and perhaps provide a justification? It would also be good to present data for repeated applications of ATP alone using protocols similar to those for testing antagonists so the reader can better appreciate the extent of run down with different recording configurations for the different receptors.

We thank the reviewer for bringing up this point. The whole-cell configuration is the most commonly used method in patch-clamp experiments; therefore, we used this method to record the current of pdP2X7 (Author response image 1). However, the whole-cell configuration is not suitable for all experiments; for example, the currents of P2X1 and P2X3 recorded by this method show a severe "rundown" effect. The "rundown" effect prevents accurate calculation of the inhibition rate of the antagonist, and to obtain more accurate results, we used perforated patches to record the currents of hP2X1 and hP2X3.

**Author response image 1. sa3fig1:** Representative current traces of pdP2X7, hP2X3, and hP2X1 after repeated applications of ATP. The pdP2X7 currents were recorded using the whole-cell configuration, and the hP2X1 and hP2X3 currents were recorded using perforated patches.

(3) The data in Fig. S1, panel A shows multiple examples where the currents activated by ATP after removal of the antagonist are considerably smaller than the initial ATP application. Is this due to rundown or incomplete antagonist unbinding? It is interesting that this wasn't observed with hP2X1 and hP2X3 even though they have a higher affinity for the antagonist. Showing examples of rundown without antagonist application would help to distinguish these distinct phenomena and it would be good for the authors to comment on this in the text. It is also curious why a previous study on pdP2X7 did not seem to have problems with rundown (see Karasawa and Kawate. eLife, 2016).

We thank the reviewer for bringing up this point. We believe that this difference may be the result of incomplete antagonist unbinding. A similar phenomenon has been observed in previous studies of pdP2X7 (eLife 2016, PMID: 27935479). In the previous experiment, the currents activated by ATP after removal of the antagonist A740003 did not return to the initial value upon ATP application, whereas activation by ATP after removal of the antagonist GW791343 immediately restored the initial value upon ATP application (Fig. 1C of eLife 2016, PMID: 27935479). This may be because different inhibitors dissociate differently from pdP2X7. In our experiments, we assumed that PPNDS/PPADS was not completely dissociated from P2X7 even after 20 min of elution. The activation of P2X7 by ATP without antagonists showed no rundown effect (Author response image 1); therefore, we calculated the inhibition rate of the antagonist according to the precontrol.

(4) The written presentation could be improved as there are many instances where the writing lacks clarity and the reader has to guess what the authors wish to communicate.

To address this comment, we made changes to the text, particularly by following the

**Recommendations for The Authors**

**Reviewer #1 (Recommendations For The Authors):**
(1) The way the manuscript is written could be greatly improved. There are many confusing sections where the reader has to guess what the authors wish to convey. For example, on page 9 "In addition, the mutation of Val173 to aspartate, as observed in pdP2X7, significantly decreased the sensitivity to PPNDS (Fig. 6B)." It appears from this sentence that Asp is present in P2X7, which is incorrect, please rephrase. There are many more examples of confusing sentences that need to be carefully edited to improve comprehension.

To address this comment, we extensively modified the text to avoid this kind of misunderstanding. Please see the manuscript file with the track changes.

(2) Please use either a 1-letter or 3-letter code for amino acid residues throughout the manuscript to maintain uniformity.

We made this correction throughout the revised manuscript.

(3) In Figure 1 on the right side, including the nearby density and side chains for interacting residues of PPNDS and PPADS would give more information and reliability for the density of the drugs.

We appreciate this comment. The corresponding information is shown in Fig. S7.

(4) Typo: Figure S1, E, and F panels - please correct the y-axis label to Inhibition.

We corrected the typo in Fig. S1.

(5) Please rewrite the legends for Fig. S3 and S5. They are confusing. The figure shows 3D classification using Relion, however, the legend suggests it was done using Cryosparc. Please clarify.

We apologize for the confusion. Before applying C3 symmetry, all steps including 3D classification were performed in Relion 3.1. With C3 symmetry, we performed further refinement using Cryosparc v4.2.1 by non-uniform refinement. We have corrected the figure legends accordingly.

(6) For Fig. S3 and S5 increase the resolution and size of representative micrographs, and also please provide scale bars.

We have corrected Figures S3 and S5 accordingly.

(7) Please add the 3D classification protocol performed in Relion/Cryosparc in the methods section as well.

We added the corresponding description to the revised manuscript (Lines 9-14, Page 16).

(8) In Table S1, under the initial model the authors state 'this study' when they should report the use of 5U1L according to the methods section.

We corrected Table S1 in accordance with this comment.

(9) The authors should consider combining the raw data shown in Figure S1 in Figure 6 as it provides stronger support for the conclusions than the bar graphs shown in Figure 6B.

We appreciate the comment and fully understand the intention of Reviewer #1. Nevertheless, we would like to keep Figure S1, since it was also mentioned earlier together with Figure 1. In addition, if we combine Figure S1 with Figure 6, the result would be too large to present as a single figure.

(10) In Figure 6A, please provide colored labels for both P2X7 and P2X1 to aid comprehension of the structural models.

Based on this comment, we corrected the labels in Figure 6.

(11) In the discussion, the authors write about comparisons with the docking study by Huo et al. JBC, 2018. Can they show the superimposition of their EM model with the previous studies' docking model in a supplementary figure for more clarity?

We appreciate the constructive comments. However, unfortunately, the docking model in the previous study (JBC 2018, PMID: 29997254) is not available, so it is not possible to show the superimposition.

**Reviewer #2 (Public Review):**
Summary:P2X receptors play pivotal roles in physiological processes such as neurotransmission and inflammation, making them promising drug targets. This study, through cryo-EM and functional experiments, reveals the structural basis of the competitive inhibition of the PPNDS and PPADS on mammalian P2X7 receptors. Key findings include the identification of the orthosteric site for these antagonists, the revelation of how PPADS/PPNDS binding impedes channel-activating conformational changes, and the pinpointing of specific residues in P2X1 and P2X3 subtypes that determine their heightened sensitivity to these antagonists. These insights present a comprehensive understanding that could guide the development of improved drugs targeting P2X receptors. This work will be a valuable addition to the field.Strengths and weaknesses:The combination of structural experiments and mutagenesis analyses offers a deeper understanding of the mechanism. While the inclusion of MD simulation is appreciated, providing more insights from the simulation might further strengthen this already compelling story.”

We appreciate the time and effort Reviewer #2 devoted to this review, and we have addressed the specific comments below.

**Reviewer #2 (Recommendations For The Authors):**
(1) On page 3, the sentence "ATP analogs are the most competitive inhibitors of P2X receptors but are typically unsuitable due to a lack of high specificity in vivo," might need additional context. Could the authors clarify if they are referring to the unsuitability of ATP analogs for medical applications?

To address this comment, we have rewritten the sentence as follows (Lines 13-16, Page 3):

ATP analogs are most common among competitive inhibitors for P2X receptors; however, they are generally unsuitable for in vivo applications due to their relatively low specificity, which may result in off-target toxicity. This issue arises because the human body contains numerous ATP-binding proteins.

(2) Fig. S1. I am curious why, for P2X7, the ATP-only current after removal of PPNDS/PPADS does not recover and become larger than the current in the presence of PPNDS/PPADS? Such behavior was not as pronounced in P2X1. Does that suggest PPNDS/PPADS might remain bound and can not be removed when the P2X7 channel is closed?

We thank the reviewer for bringing up this point. We believe that this difference may be the result of incomplete antagonist unbinding. A similar phenomenon has been observed in previous studies of pdP2X7 (eLife 2016, PMID: 27935479). In the previous experiment, the currents activated by ATP after removal of the antagonist A740003 did not return to the initial value upon ATP application, whereas activation by ATP after removal of the antagonist GW791343 immediately restored the initial value upon ATP application (Fig. 1C of eLife 2016, PMID: 27935479). We strongly agree with the reviewer that this may be due to the difficulty of dissociating the antagonist from pdP2X7.